# Addressing Sexually Transmitted Infections Due to *Neisseria gonorrhoeae* in the Present and Future

**DOI:** 10.3390/microorganisms12050884

**Published:** 2024-04-28

**Authors:** Julia Colón Pérez, Rosa-Antía Villarino Fernández, Adrián Domínguez Lago, María Mercedes Treviño Castellano, María Luisa Pérez del Molino Bernal, Sandra Sánchez Poza, Eva Torres-Sangiao

**Affiliations:** 1Servicio de Microbiología y Parasitología Clínica, Complexo Hospitalario Universitario de Santiago de Compostela, 15706 Santiago de Compostela, Spain; julia.colon.perez@sergas.es (J.C.P.); adrian.dominguez.lago@sergas.es (A.D.L.); maria.mercedes.trevino.castellanos@sergas.es (M.M.T.C.); maria.luisa.perez.del.molino.bernal@sergas.es (M.L.P.d.M.B.); 2Grupo Microbiología, Instituto de Investigación Sanitaria de Santiago de Compostela (IDIS), 15706 Santiago de Compostela, Spain; 3Departamento de Microbiología, Facultad de Farmacia, Universidad de Santiago de Compostela, 15782 Santiago de Compostela, Spain; rosaantia.villarino@rai.usc

**Keywords:** neisseria gonorrhoeae, sexually transmitted diseases, cephalosporin resistance, virulence, vaccine

## Abstract

It was in the 1800s when the first public publications about the infection and treatment of gonorrhoea were released. However, the first prevention programmes were only published a hundred years later. In the 1940s, the concept of vaccination was introduced into clinical prevention programmes to address early sulphonamide resistance. Since then, tons of publications on *Neisseria gonorrhoeae* are undisputed, around 30,000 publications today. Currently, the situation seems to be just as it was in the last century, nothing has changed or improved. So, what are we doing wrong? And more importantly, what might we do? The review presented here aims to review the current situation regarding the resistance mechanisms, prevention programmes, treatments, and vaccines, with the challenge of better understanding this special pathogen. The authors have reviewed the last five years of advancements, knowledge, and perspectives for addressing the *Neisseria gonorrhoeae* issue, focusing on new therapeutic alternatives.

## 1. Introduction

*Neisseria gonorrhoeae* (NG) is a Gram-negative diplococcus that infects the urogenital, rectal, and pharyngeal areas in both men and women, who experience purulent urethral discharge and dysuria, respectively. Usually, most of cases are asymptomatic [1], but untreated infections can cause severe complications, ranging from epididymitis and salpingitis to pelvic inflammatory disease (PID), ectopic pregnancy, and infertility, as well as newborn blindness. In fact, *N. gonorrhoeae* (NG) is one of the major public health threats worldwide with 87 million new cases in 2016, an increased transmission from 78 million in 2012 [2,3]. NG infection or gonorrhoea is a sexually transmitted infection (STI), the second leading bacterial STI around the world after *Chlamydia trachomatis* [4,5].

In 2020, the Centre for Disease Control and Prevention (CDC), reported a total of 677,769 cases of gonorrhoea (an increase of 111% since 2009) [6], and around 1.57 million cases of incident gonorrhoea infections were estimated to occur annually in the United States [7]. More than half of the gonococcal isolates in 2020 were resistant to at least one class of antibiotics [8].

An *N. gonorrhoeae* diagnosis should be motivated by the observation of specific clinical signs or scheduled by regular screenings in high-risk groups. The gonorrhoeae screening is the main key in detecting asymptomatic infections as well as co-infections, which are common for *C. trachomatis*. The current gold-standard for NG diagnosis is nucleic acid amplification tests (NAATs), yet [9] NAATs are the recommended diagnostic tools for NG diagnosis in urethral, pharyngeal, and rectal samples, because of their high sensitivity and specificity [10]. In fact, NG NAATs are usually incorporated as multiplex panels with syndromic tests to address the diagnosis of STIs, including pathogens other than NG, such as *C. trachomatis*, *T. pallidum*, *M. genitalium*, and *T. vaginalis* among others [11]. In addition, molecular assays are good candidates for point-of-care tests, shortening turnaround time and allowing for proper treatment at the soonest, which has a great impact on transmission and the emergence of resistance. However, one limitation of NAATs as a daily routine to detect AMR determinants is their high cost. Therefore, conventional culture is still crucial to evaluate NG resistances and must be included in diagnostic procedures [12].

Certainly, one of the major concerns is the increase in NG infections observed globally and the emergence of antimicrobial resistance [13]. The current dual treatment is with ceftriaxone and azithromycin, or a single dose of ceftriaxone or cefixime [14,15]; however, resistance to both ceftriaxone and azithromycin have already been identified [16,17]. The impact on commensal organisms [18] with the few alternative antibiotics in the pipeline underscores the critical need for new public health strategies for gonorrhoea prevention and control. In fact, the WHO lists NG as a “priority pathogen” for which new therapies are urgently needed [19]. It is essential to match the available antimicrobial therapies, discover new alternative treatments, and develop vaccines to contain both the high prevalence and growing resistance of NG, as well as better programmes for prevention.

At present, there are limited available alternative regimens reliable enough for gonorrhoea treatment. Moreover, the need to improve the prevention programmes, especially for high-risk population, such as men who have sex with men (MSM). MSM are disproportionally affected by STIs [20] and are often asymptomatic, therefore their condition remains frequently undetected and untreated [21], leading to severe sequelae and serving as a reservoir for continued transmission. Extragenital STIs in MSM are frequent [20,21] and can contribute substantially to the further spread if not diagnosed and treated. This population is the best candidate for vaccination because it could also potentially reduce antimicrobial resistance.

In this review, we will discuss the current situation of NG resistance and the advancements in new treatments, with a special focus on novel vaccine development, as well as improved prevention programmes. All information has been gathered from the relevant articles in PubMed and OVID Medline, under the search terms “*Neisseria gonorrhoeae*” and “antimicrobial resistance” or “epidemiology” or “prevention” or “treatment” or “vaccines” or “virulence factors” or “omics” published from 2016 to 2023. The selection was made based on the impact factor of the journal (more than 4) as well as relevance and citations.

## 2. Virulence Factors of *Neisseria gonorrhoeae*

*N. gonorrhoeae* is a well-known human pathogen that colonises the genitourinary tract and which diverges significantly from another commensal *Neisseria* spp. The infection of the lower genitourinary tract mucosa leads to urethritis in males and cervicitis in females, including inflammation with a neutrophilic purulent exudate. It is estimated that 50% of gonococcal infections in women are asymptomatic compared to a much lower proportion in men. Asymptomatic infection has a vague pathological definition and is almost recognisable through the recovery of viable gonococci from an asymptomatic patient, usually during a routine STI screening [22]. Different clinical manifestations induced by the gonococcus range from pelvic inflammatory disease and/or a disseminated infection to pharyngitis and/or asymptomatic infections. 

The mechanisms that facilitate their ability to survive and persist in different anatomical sites are a combination of mechanical routes of entry and genetic determinants. NG has multiple virulent factors that assist in avoiding the immune system by utilising various mechanisms. Certain factors can display antigenic and phase variability, which make them undetectable by immune cells. Other factors help in functions such as adhesion, invasion, colonisation, and nutrient acquisition. Refer to Table 1 for a list of the main virulence factors.

## 3. Antimicrobial Resistance in *Neisseria gonorrhoeae*: Where We Are?

During the 20th century, successful antimicrobial treatments for infectious diseases were reached, especially from the mid-1940s to 1970s, the golden era of antimicrobial discovery. Nevertheless, antimicrobial resistance (AMR) began to substantially menace treatments and the control of many infectious diseases, prompting the development and introduction of new antimicrobials. Indeed, pathogens have continued to develop AMR mechanisms faster than new therapies have been developed, and during the 21st century, only a few antimicrobials with new mechanisms of action have been developed [55] (Figure 1).

Sulphonamides, penicillin, or tetracyclines initially reported satisfactory results for NG treatment. But in a short lapse of time, they were rendered useless because of the spread of resistant strains. Other antibiotics like macrolides or cephalosporins remain reliable in most cases, provided some of them are not to be used in monotherapy to avoid resistance development. The current recommended first-line treatments for gonorrhoea consist of ceftriaxone as a single-drug therapy, or a dual therapy combining ceftriaxone and azithromycin. However, a decrease in susceptibility to both these drugs and subsequent treatment failures are being reported in many countries, resulting in rising concern [55] (Figure 1).

In addition, NG MDR strains are described as those that exhibit resistance to one class of the generally recommended antibiotics, which are the following: cephalosporins (oral and injectable ones are considered separately) and spectinomycin; and two or more of the less frequently used classes, which are the following: penicillin, fluoroquinolones, azithromycin, aminoglycosides, and carbapenems. XDR strains are resistant to ≥2 classes of the first group and to ≥3 classes of the second one. Unfortunately, NG is evolving into a superbug, which is a major public health concern around the world. The present situation has stemmed from the high rate of antimicrobial usage, suboptimal control and monitoring of AMR, and treatment failures, plus the extraordinary capacity of NG to develop and retain AMR, emerging as a silent epidemic [55].

Antimicrobial resistance for each drug is caused by the following resistance determinants: *folP*—sulfonamide resistance; *penA*, (*mtrR* + *penB* + *ponA*)—chromosomal penicillin resistance; 16S rRNA mutations, *rpsE*—spectinomycin resistance; *rpsJ*, (*mtrR* + *penB*)—chromosomal tetracycline resistance; *bla_TEM-1_*, *bla_TEM-135_*—plasmid-mediated penicillin resistance; *tetM*—plasmid-mediated tetracycline resistance; *gyrA*, *parC*—fluoroquinolone resistance; *penA* mosaic allele—cefixime resistance; 23S rRNA mutations, (*erm* + *mefA*)—azithromycin resistance; novel *penA* alleles—ceftriaxone resistance.

SUL, sulfonamides; PEN, penicillin; SPT, spectinomycin; TET, tetracycline; CIP, ciprofloxacin, OFX, ofloxacin; CFM, cefixime; CRO, ceftriaxone; AZM, azithromycin; DOX, doxycycline [55].

### 3.1. Mechanisms of Resistance: Overview

Briefly, NG presents all the following resistance mechanisms characterised to date: enzymatic destruction, target modification, decreased influx of antimicrobials due to loss of porins and increased efflux of antimicrobials due to pump production.

The main mechanism behind penicillin and extended spectrum cephalosporin (ESC) resistance is mediated by chromosomal mutations in *penA* and *ponA*. Mutations in *mtrR*, *porB*, and *pilQ* also play a significant role. Point mutations in PBP proteins (PBP1 in the case of *ponA* and PBP2 in the case of *penA*), *penA* mosaic alleles, and single nucleotide polymorphisms (SNP) in *ponA* determine high-level penicillin resistance [56]. The mosaic region of *penA* is composed of DNA sequences inserted from commensal *Neisseria* transformation (the horizontal transfer of partial or whole genes) [55,57]. These acquired mosaic alleles are the main cause of cephalosporin resistance as well and are especially associated with higher cefixime minimum inhibitory concentrations (MICs). Moreover, specific SNPs are statistically associated with higher cephalosporin MICs, but the underlying mechanisms have yet to be elucidated [55,57]. 

Other SNPs in *porB*, which encode the outer membrane porin PorB, decrease the influx of penicillin and tetracyclines into the periplasm [56]. Missense mutations in *pilQ2*, which encodes a pore-forming secretin, lead to a similar outcome because of the destabilisation of pore formation around the pilus [56]. High-level spectinomycin resistance in NG can be caused by SNPs in the 16S rRNA binding region and by deletions in the 30S ribosomal protein S5, encoded by the *rpsE* gene [55].

Regarding quinolone resistance, it can occur due to specific SNPs in *gyrA*, resulting in low- and intermediate-level resistance. Higher levels of resistance require additional specific mutations in *parC* that develop when exposed to subinhibitory ciprofloxacin concentrations. Nonetheless, mutations in *gyrB* and *parE* do not seem to impact ciprofloxacin resistance [55].

Essentially, *erm* genes can cause high-level resistance to erythromycin and low-level resistance to azithromycin, but they are not the main source of macrolide resistance. Rather, most macrolide-resistant strains present mutations in the 23S rRNA, where the number of mutated alleles is what determines how high the macrolide MICs are. The number of mutated alleles can accumulate rapidly when exposed to subinhibitory conditions [50]. Notably, the internationally spreading multi-antigen sequence typing (NG-MAST) genogroup G12302 contains four mutated alleles coupled with mosaic *mtrD* and *mtrR* promoter mutations [55,58]. 

A deletion in the promoter region of the repressor protein MtrR results in an overexpression of the Mtr–CDE efflux pump. This mechanism affects several antibiotics such as penicillin, cephalosporins, tetracyclines, and macrolides, as well as host-derived antimicrobial compounds like fatty acids, bile and even cathelicidin LL37, which can enhance biological fitness [55]. 

On the other hand, plasmid-mediated resistance to penicillin in NG is mediated via the production of penicillinase. These strains carry the *bla_TEM_* gene in an acquired plasmid from *H. parainfluenzae* [56]. Another plasmid-mediated AMR determinant is *tetM*, which confers high-level resistance to tetracyclines and is possibly acquired from *H. influenzae* [55]. The first β-lactamase identified in NG strains that exhibited high-level resistance to penicillin without altering ESC MICs was TEM-1. This β-lactamase differs from TEM-135 by a single SNP. While TEM-135 does not seem to increase NG MIC to ESCs (albeit it does increase penicillin MIC), its sequence only requires one additional SNP to become a TEM-type extended-spectrum β-lactamase (ESBL), such as TEM-20, TEM-106, or TEM-126 [59,60]. The spread of an ESBL capable of degrading ceftriaxone could render gonorrhoea as an untreatable disease in most settings worldwide, where ceftriaxone is the last remaining option for empiric first-line antimicrobial monotherapy [57].

Figure 2 shows the main mechanisms of resistance of NG as follows: (1) Penicillin-binding protein (PBP) mutations in beta-lactam and cephalosporin resistance; (2) Gene mutations encoding porin in beta-lactam, cephalosporin, and tetracycline resistance; (3) Overexpression of Mtr–CDE pumps increase antimicrobials’ efflux such as beta-lactams, cephalosporins, and tetracyclines. The repressor gene mutation, alongside the upregulation of *mef*- and *mac*AB-encoded efflux pumps, increases macrolides efflux; (4) Hydrolysation of the ß-lactam ring by plasmid-encoded penicillinase in beta-lactams; (5) A single nucleotide polymorphism (SNP) mutation in the 30S ribosomal protein (*rps*J) and plasmid-encoded TetM protein lower the tetracycline affinity to a ribosome; (6) 23S rRNA SNP mutation and methylation by rRNA methylases (encoded by *erm*) block the macrolides from binding to the ribosome; (7) An SNP mutation in 16S rRNA and *rps*E encoding 30S ribosomal protein S5 mutant (RPS5) inhibit spectinomycin binding to the ribosome; (8) An SNP in DNA gyrase and topoisomerase reduces the binding of fluoroquinolones to these enzymes for DNA synthesis blocking; (9) DHPS-encoding *fol*P mutations and overproduction of PABA impair sulfonamide effectivity and increases production of tetrahydrofolate. In addition, there is a figure showing the main mutations for *gyr*A, *pen*AB and *mtr*R genes (Images modified by the author) (Habiburrahman, M. et al. (2020), Copyright (c) 2020 Indonesian Journal of Pharmacy, under a Creative Commons Attribution-ShareAlike 4.0 International License) [56].

### 3.2. Mechanisms of Resistance: Spreading and Fitness

The prevalence of resistant strains to most antibiotics available is high and rising mainly because of two reasons. First, NG is naturally competent for transformation with the DNA of non-pathogenic *Neisseria* [61]. Transformation or horizontal gene transfer is an important mechanism of genetic diversity that allows NG to adapt rapidly and survive in hostile environments [55,61]. The combination of similar genes via transformation may produce mosaic genes that represent resistance determinants, such as in the case for *penA* [57]. Secondly, there are several commensal *Neisseria* species that subsist in the pharynx ecosystem, acting as the reservoirs of resistance-coding genes. These genes can be easily transferred to pathogenic *Neisseria* in the context of an asymptomatic pharyngeal infection, or even during a temporary colonisation after oral sex [55,61,62]. It is in pharyngeal infections where most treatment failures have been reported, mostly associated with MSM [63]. Moreover, evidence suggests this has been the case for the spread of *penA*, *mtr*, and *gyrA* mutations [64]. 

Usually in nature, resistant strains display advantages when exposed to antibiotic pressure, but their growth is compromised under normal conditions in the absence of compensatory mutations. Nonetheless, there is evidence of NG resistant strains that seem to display improved biological fitness [65]. Goytia et al. [64] proposed a different approach regarding the management of NG resistance that involves the specific study of commensal *Neisseria* and the quantification of transformation rates to identify new treatments and vaccine targets [64]. 

Thus, multiple research groups have focused on the study of these fitness costs. D’Ambrozio et al. found that a specific mutation in *gyrA* conferred a fitness advantage in vivo because of its effects on genome regulation [61,66]. These findings are supported by the fact that fluoroquinolone-resistant NG strains remain prevalent even after fluoroquinolones were discarded as primary treatment options. However, they observed that a second-stage mutation in *parC* negatively affected biological fitness [61]. Likewise, mutations that affect the Mtr–CDE efflux pump, allowing the excretion of antibacterial peptides, have proven to be advantageous for competitive growth in vivo [55,61].

Controlling the spread of these AMR is therefore key to protecting the current ceftriaxone–azithromycin dual therapy [67]. Luckily, concomitant resistance to both azithromycin and cephalosporines remains extremely rare internationally, so the few ceftriaxone-resistant cases could be treated with azithromycin [67]. Further genomic surveillance studies are needed to elucidate how the accumulation of resistance determinants impacts biological fitness and lineage spread [68].

### 3.3. Mechanism of Resistance: XDRs and Clones

The first XDR NG strain displaying high-level resistance to cefixime and ceftriaxone, H041, was identified from a pharyngeal sample of a female sex worker in 2009 in Japan. This strain carries a mosaic *penA_H041_* allele responsible for cephalosporin resistance as well as other resistance determinants (*mtrR*, *penB*, *ponA1*). It has been suggested that this mosaic allele was transferred from commensal *Neisseria* to NG in a pharyngeal infection [61]. Evidence revealed that H041 is a subclone of the internationally spreading cefixime-resistant NG-MLST ST7363 (novel NG-MAST ST4220), proving that NG can develop resistance to ceftriaxone [13,69]. In addition, Golparian et al. suggested that NG strains can develop ceftriaxone resistance through a single horizontal gene transfer, which raises great concern [70].

The second NG strain showing high-level resistance to cefixime and ceftriaxone, F89, was identified from an MSM in France in 2010. F89 is a subclone of NG-MLST ST1901 (NG-MAST ST1407) and likely originated in Japan. Unemo et al. conducted transformation experiments with reference strains from the WHO collection and observed that the novel *penA* mosaic allele was the primary cause of ESC resistance. However, high ESC rates (MIC = 4 μg/mL and MIC = 1 to 2 μg/mL for cefixime and ceftriaxone, respectively) were motivated by the synergistic effects of additional resistance determinants, such as *mtrR* and *penB*. F89 is considered XDR as it is also resistant to fluoroquinolones, macrolides, tetracycline, trimethoprim–sulfamethoxazole, and chloramphenicol [71].

The first two ESC-resistant and MDR NG in Spain were detected in two sexually related MSM, being the first documented case of inter-patient transmission of ceftriaxone-resistant NG. NG-MAST showed both isolates belonged to ST1407, like F89 [72]. The ST1407 lineage is associated with ESC resistance and MSM in the EU and EEA. Fortunately, there has been a reduction in its spread since 2009 and 2010, corresponding with a rising tendency in cephalosporine susceptibility [67].

Both high-level ceftriaxone resistant strains, H041 and F89, were first identified in high-risk groups (sex workers and MSM), where transmission often occurs [61]. Although this causes great concern, it is also worth bearing in mind that these *penA* alleles reduce biological fitness, thus limiting the spread, considering this gene is involved in cell-wall biosynthesis. This idea is further supported by the fact that high-level ceftriaxone resistance remains extremely rare. Indeed, no additional XDR ceftriaxone-resistant strains have been described to date [68]. This could be either explained because of the recent introduction of ESCs as first-line drug in treatment guides or due to biological fitness loss [72]. 

Numerous cases of ceftriaxone treatment failures in recent years have been linked to the ceftriaxone-resistant FC428 clone, which was first identified in 2015 in Japan [70,73]. This clone carries the mosaic *penA* allele, 60.001, which contains the A311V and T483S polymorphisms, previously identified in high-level ceftriaxone-resistant strains like H041 or F89, among others. Kanesaka et al. found evidence suggesting that the mosaic *penA* allele in FC428 is derived from the ceftriaxone-resistant *N. subflava* [61]. Furthermore, NG FC428 also presents resistance to spectinomycin and azithromycin and its identification has since spread internationally [61].

Additionally, a sub-lineage of ST7363 that lost the mosaic *penA* allele through recombination was found in a genomic surveillance study conducted by Yahara et al. This sub-lineage also results in a deletion in the *mtrR* promoter that predicts azithromycin resistance. These findings suggest that loss of AMR could be beneficial for NG in the absence of antibiotic pressure [68,73]. Furthermore, Sánchez-Busó et al. defined a novel NG-MAST genogroup, G12302, which carries a *N. lactamica*-like mosaic *mtrR* promoter and a *mtrD* sequence [67]. This lineage is strongly associated with pharyngeal infections in MSM and is primarily responsible for the rise in low-level resistance to azithromycin in Europe.

### 3.4. Resistance-Guided Therapy

Resistance-guided therapy presents as an interesting approach if pathogen detection and molecular resistance assays are coupled. For example, for ciprofloxacin, the mechanism of resistance is relatively simple—by the detection of a single mutation at the serine 91 codon of the *gyrA* gene. However, there is no single marker that can predict reduced susceptibility to ESCs and azithromycin [63]. Anyhow, some AMR alone are sufficient to cause treatment failure. Most often, the clinical outcome responds to the accumulative effect of retaining several AMR and epistatic interactions [55,64]. 

Regarding fluoroquinolones, *gyrA* genotyping in conjunction with pathogen detection proves cost-efficient only in areas where ciprofloxacin susceptibility is prevalent [57]. A shift in favour of ciprofloxacin prescription would slow the rate of ESC resistance, reduce costs when compared to ceftriaxone treatments, while also sharing with cefixime the advantage of not requiring clinic visits for injection [57]. In a recent study, Trick et al. developed a portable, rapid, on-cartridge magneto fluidic purification and testing (PROMPT) PCR test that could simultaneously detect and genotype NG to predict ciprofloxacin resistance (assessing *gyrA* and *opa* genes) in less than 15 min. These assays proved to be highly sensitive and specific but lacked information regarding other STIs. Future developments would benefit from implementing multiplex assays, including another AMR biomarkers or other prevalent STIs like chlamydia [74].

Although ceftriaxone has proven to be the best performing of the tested injectable drugs in a previous meta-analysis, correspondingly, azithromycin was the best oral option [75,76]. Treatment failures with the standard ceftriaxone dosage are mostly observed in pharyngeal infections, mainly due to pharmacokinetic and pharmacodynamic reasons. In these situations, a higher dose may be needed to achieve a longer exposure above the MIC. In recent trials ertapenem has been proposed as an alternative in these situations, but it should not be applied to first-line options to preserve it for other infections caused by MDR Gram-negative bacteria [9,14].

Antibiotic stewardship is suggested as a powerful tool to decrease antibiotic pressure, when possible, decreasing microbiota exposure and resistance selection [18]. Genomic surveillance via whole-genome sequencing (WGS) in conjunction with epidemiological and AMR data is vital to identify AMR prevalence and transmission of gonococcal lineages, and to reduce the incidence of infections and antimicrobial resistance in NG. Therefore, the emergence of multidrug-resistant strains of NG demands new protocol implementations to prevent its dissemination among the population. 

## 4. Prevention and Plans of Action to Overcome *Neisseria gonorrhoeae*

Prevention of gonorrhoea has relied on public health measures, such as condom use, educational messaging, or screening to detect asymptomatic carriage [77]. In confirmed cases, early treatment of symptomatic and asymptomatic cases is essential, continuing with the exhaustive retrospective follow-up of sexual contacts to break the chain of transmission. The optimal management of gonococcal infection involves synergistic actions, namely between the identification and detection of AMR, new therapeutic drugs, and candidate vaccines, supported by proper prevention programmes. Despite these measures, the incidence of gonorrhoea is increasing, predominantly in MSM, but also more recently in heterosexual populations [78,79]. Therefore, sexual risk behaviours are the key point to tackle, trying to reduce them as efficiently as possible, strongly advising the use of condoms (the barrier method par excellence to avoid the transmission), as well as routine check-ups [77].

Only 10% of heterosexual men use it in the sexual sphere while in MSM this percentage doubles, showing that its use is scarce in all types of relationships [80]. The reasons are unclear, and likely support that using online dating applications can be associated with outbreaks of bacterial STIs among MSM [81,82]. In fact, MSM are disproportionally affected by STIs, being the major focus of VIH pre-exposure prophylaxis (PrEP) programmes. The introduction of PrEp programmes has been associated with a decrease in condom use and consequently an increase in STIs [83,84]. Additionally, among the reported PrEP users, chemsex has been linked to an increased incidence of gonorrhoea and chlamydia. This effect has been stronger for people reporting multiple chemsex substances, highlighting the need for integrated services that address the complexities of sexualised substance use. Deborah A. Williamson et al. [77] demonstrated the transmission and spread of gonococcal lineages within and across distinct sexual networks. The authors also identified several potential touchpoints that promoted the dissemination of NG, namely PrEP use, oropharyngeal gonorrhoea in female sex workers, returning international travellers, and MSMW who may facilitate the bridge between MSMO (only men) and heterosexuals.

The collectives with high-risk sexual behaviours have a higher likelihood of being infected with other STIs. Most of them are patients of PrEP and PEP (Post Exposure Prophylaxis) for the prevention of HIV infection, who have exponentially increased over the last decade. Several studies have been conducted incorporating doxycycline as prophylaxis against other STIs (doxy-PrEP and doxy-PEP) [85] because of its high activity against *C. trachomatis* and *T. pallidum* and low resistance shown, regardless of the well-established resistance patterns for NG and *M. genitalium* [86]. Although doxy-PEP can be an excellent STI prevention strategy, more studies are necessary to investigate the long-term impact on resistance profiles [87,88].

On the other hand, increasing travel is leading to the increased importation of NG (particularly AMR strains) from areas with a high prevalence of STIs, with subsequent endemic local transmission [89,90]. Preventing the emergence and spread of AMR in NG is imperative, so disease control centres’ response programmes should be monitored regularly to promptly identify and address areas for improvement. The effectiveness and control of AMR in NG need strong support from comprehensive management and control strategies, nationally and internationally, including the following: (i) appropriate STI prevention, e.g., promotion of condom use, (ii) diagnostic and testing algorithms, e.g., triple-site testing in men who have sex with men, (iii) treatment, (iv) test of cure, (v) notification and treatment of partners, and (vi) robust epidemiological surveillance to identify key groups at risk of gonorrhoea and gonococcal AMR [91].

Treatment guidelines for gonorrhoea usually give a mono-species approach, considering the AMR-inducing effects on NG. Otherwise, Kenyon et al. [92] proposed a “pan-Neisseria” plan of action, taking into consideration the commensal *Neisseria* involved in NG reinfection via horizontal gene transfer. However. dual therapy is expected to eradicate NG more efficiently than monotherapy, despite a negative effect on commensals, important constituents of a healthy microbiome and transfer resistance determinants [64,92]. Resistance-guided therapies are directly conditioned by antimicrobial resistance prevalence, and this dictates how effective and cost-efficient they would be in reducing antibiotic pressure [57]. In areas where the prevalence of resistance to ciprofloxacin is low, the implementation of *gyrA* genotyping in conjunction with pathogen detection is proving to be cost-effective. Such efforts can be facilitated by the incorporation of resistance marker determination into molecular point-of-care tests [57,93,94]. Unfortunately, the data to date are limited and further studies are needed to assess the applicability of resistance-guided therapy [57,93,94].

On the other hand, it is well established that early testing and diagnosis are vital to stopping the spread of STIs, as highlighted during the pandemic and the Monkey Pox outbreak. Certainly, in countries with good surveillance systems, this is manageable, but it represents a major challenge in low- and middle-income countries. In these scenarios, the development of accessible, low-cost point-of-care testing for STIs would reduce healthcare costs and treatment failures, as well as slow the emergence of antimicrobial resistance [95,96].

The Global Health Sector Strategy on HIV, Hepatitis, and STIs (2022–2030) aims to reduce the incidence of new cases of gonorrhoea in people aged 15–49 by 90% in 2030, from 82.3 million/year in 2020 to 8.23 million/year. The goal requires the following two plans of action: AMR and the control of gonorrhoea; however, effective vaccines remain the best strategy. The current WHO strategy focuses on the surveillance of antimicrobial resistance in gonorrhoea through the Enhanced Gonococcal Antimicrobial Surveillance Programme (EGASP) to ensure quality data that are useful for treatment recommendations and policies [97,98].

In summary, the measures offered by the governmental and non-governmental organisations in this field are as follows [99]:-Improved case reporting systems, allowing more real prevalence figures to be known;-Treatment regimens that ensure greater patient adherence;-Organisation of sex education and information programmes on STIs that are oriented and easily accessible to most of the population, especially in schools and high schools.

## 5. Vaccine Development, the Future?

The high incidence of *N. gonorrhoeae*, its high morbidity, and the worrying increase in strains resistant to multiple antibiotics have accelerated efforts to develop an effective vaccine against gonococcus. There is no single consistent class of antimicrobials appropriate for the treatment of NG, even the dual treatment approaches. Gonococcal infections can persist and reinfect the host on the basis that this bacterium can dodge and overwhelm the immune responses of the individual [100]. Therefore, gonococcal vaccines may ultimate the prevention of adverse outcomes and reduce the impact of gonococcal resistance.

*Neisseria meningitidis* can be characterised into 12 serogroups, mostly common A, B, C, W135, X, and Y serogroups [101]. Serogroup B (MenB) has been predominant in children and young adults, and broadly diffused, pushing the first licensed vaccine in 2015 available for people aged 12–25 years. The 4CMenB (Bexsero^®^, GSK, Siena, Italy, vaccine is a four-component vaccine based on recombinant proteins of the pathogen strain MC58. The vaccine contains two fusion proteins, namely the Neisserial heparin binding antigen-GNA1030 (NHBA, peptide 2) and the factor H binding protein-GNA2091 (fHbp, peptide 1, subfamily B), and the single antigen Neisserial adhesin A (NadA, peptide 8), combined with the outer membrane vesicles (OMV) [102]. Furthermore, rLP2086 (Trumenba^®^, Pfizer, Brussels, Belgium) is a bivalent vaccine containing two fHbp peptides, one from each of the two subfamilies, peptide 45, subfamily A, and peptide 55, subfamily B [103].

Interestingly, the only vaccines offered so far that appear to offer protection are the OMV vaccines against serogroup B of *N. meningitidis* (MeNZB^™^, Novartis, Basel, Switzerland [104], which are no longer available, 4CMenB [8,105,106] and VA-MENIGOC-BC^®^, Finlay Institute, Havava, Cuba [107,108,109]). Although they have moderate efficacy (31% in the case of MeNZB), mathematical models predict that they could have a significant impact on the prevalence of the disease in the population [110]; indeed, it has been observed that vaccination with these vaccines leads to a reduction in hospitalisation rates [111] and that they have a herd effect in the unvaccinated population [112]. As of 2019, new clinical trials are underway to evaluate the response to serotype B meningococcal vaccines against gonorrhoea [113].

Clinical trials of two gonococcal vaccines developed in the late 20th century were disappointing. One consisted of killed whole cells [114] and the other of purified pilus [115] Both failed to induce protection against reinfection by heterologous strains, despite generating high antibody responses. The failure of these vaccines was attributed not only to the ability of NG to evade the immune response but also to RmpM-induced antibodies blocking the formation of the complement membrane attack complex [116] or antigenic variation. 

Since then, advances in antigen purification and whole genome sequencing, the development of new proteomic, immunoproteomic, and bioinformatic techniques, the incorporation of an AI model (EDEN) [117] in the identification of protective antigens, as well as advances in the understanding of how the gonococcus is able to evade the immune response, have led to the proposal and investigation of a large pool of vaccine candidates, summarised in several reviews [106,113,118,119,120,121,122]. The different approaches include the following vaccines: based on inactivated whole cells [123]; based on Nm OMV and Ng OMV, which require adjuvants that overcome *N. gonorrhoeae*-mediated immunosuppression such as IL-12 [124] (an inflammatory cytokine that stimulates Th-1-associated immunity and potentiates humoral or antibody-mediated immunity); in protein subunits involved in adhesion and invasion, such as PilQ, Opa, OpcA, OmpA, PorB, and NHBA [125]; based on nutrient acquisition and metabolisms, such as Tbps, Lbps, ZnuD, MtrE, MetQ, AniA, and phospholipase D; in the membrane biogenesis and LOS, such as BamA and LptD; or in immune evasions such as MtrCDE, SliC, PoB, ACP, NspA, MsrA/B [126], and MIP [127]. Other approaches include chimeric antigen vaccines [121,128], epitope vaccines such as LOS-derived 2C7 epitope-based vaccines [129,130], DNA [131] or mRNA vaccines.

In addition, the approaches used in developing gonococcal vaccines in terms of vaccine delivery systems are noteworthy. For example, the use of viral derivatives that release antigens, either as non-replicating viral-like particles (VLP) or as replicating viral particles with a single replication cycle (VRP), as well as VLP-based COVID-19 vaccines, have been used to study candidates such as the following: SliC, human lysozyme inhibitor antigen [132]; PorB [133]; TbpB [134]; filamentous phage as the phage NgoΦ6 and its phagemid derivatives [135]; microparticles such as whole-cell vaccines [123,133,136,137]; protein scaffolds [138,139]; microarray patches [123]; and liposomal preparations [140]. Another interesting delivery system is that of bacterial ghosts, empty Gram-negative bacterial envelopes that retain the functional and antigenic characteristics of the envelope and can be vaccines that induce a response against the antigens of which they are composed, or vehicles with an adjuvant effect for DNA vaccines, or even drugs. Jiao et al. have used this strategy to develop vaccines based on the PorB [141] and NspA [131] antigens.

While there is hope that a gonococcal vaccine can be developed soon, several challenges need to be addressed. The most significant challenge is that humans are the only natural reservoir of gonococci, which limits the development of effective vaccines. Although female mouse models treated with 17-β-estradiol have been developed to test the efficacy of in vivo formulations, they cannot fully mimic human infection and disease. Although there is a model of controlled infection in humans, it is limited to male volunteers whose infection occurs in the urethra and may not reflect the infection in women, where the disease has its most severe consequences. For safety reasons, the infection must be treated before 6 days. Furthermore, the need for and type of immunomodulatory adjuvants must be carefully evaluated [136,142], especially in NG OMV vaccines, due to the gonococcus’ ability to evade the immune system using numerous immunosuppressive alternatives.

## 6. New Treatments and New Alternatives

The emergence of NG strains resistant to the different antimicrobials currently used in its treatment implies the need to incorporate new molecules into the therapeutic arsenal directed against urogenital gonorrhoea, despite all that this entails. The development of new drugs focused on pathologies whose treatment is short-lived is not attractive to the pharmaceutical industry, mainly for economic reasons, unlike those aimed at chronic diseases. Moreover, the rapid emergence of resistance to these drugs would mean a loss of efficacy and therefore a new search for new molecules [89,143,144]. To date, numerous molecules with potential activity against strains of NG have undergone different phases of study to prove their efficacy. However, not all of them reached advanced stages of study and only a few have been successful, including the following three different antimicrobial agents: zoliflodacin, gepotidacin, and solithromycin.

Gepotidacin (triazaacenaftilene) and Zoliflodacin (spiropyrimidintrione) are two bactericidal antibiotics targeting type II topoisomerases (DNA gyrase and topoisomerase IV) and solithromycin is an agent belonging to the fluorketolide group with bacteriostatic activity that interferes with protein synthesis. Other compounds also have antibacterial activity, such as corallopyronin A (RNA polymerase inhibitor) and the ionophore PBT2, (regulator of metal homeostasis) whose study both in vitro and in vivo may represent a great advancement in the treatment of NG MDR infections.

### 6.1. New Antimicrobials

**Zoliflodacin** is a new oral antibiotic belonging to the spiropyrimidintriones group, first in its class, and not yet commercialised. This antimicrobial agent acts as a class II topoisomerase inhibitor, specifically on the GyrB subunit, which confers bactericidal activity [145] on different microorganisms, including multidrug-resistant strains of NG [146]. Zoliflodacin has undergone phase I and phase II randomised clinical trials (RCTs). Phase I has evaluated the pharmacokinetics [147], and subsequently in phase II, the efficacy and safety in individuals of both sexes with uncomplicated gonorrhoea. The following two groups of patients were established: those treated with single dose oral zoliflodacin versus those treated with single dose intramuscular ceftriaxone [148]. A multicentre phase III RCT, referred to as Clinical Trial NCT03959527, is currently underway in several US states, two European countries (Belgium and the Netherlands), one African country (South Africa) and one Asian country (Thailand), evaluating treatment efficacy in uncomplicated gonorrhoea cases, and comparing a single oral dose of zoliflodacin versus combination therapy between a single oral dose of azithromycin and a single intramuscular dose of ceftriaxone [149]. 

**Gepotidacin** is another new drug for uncomplicated urogenital gonorrhoea. This antimicrobial agent belongs to the triazaacenaftilene group, being the first representative of this group without marketing authorisation yet. Its mechanism of action is at the topoisomerases level, specifically class IIA topoisomerases, selectively inhibiting the GyrB subunit of DNA gyrase and topoisomerase IV. Enzymatic inhibition at the chromosome level causes bacterial cell death, hence it is classified as a bactericidal agent [150,151]. Like other drugs developed for therapeutic purposes, gepotidacin has undergone several phases of randomised clinical trials. Phase II trial has shown that a single dose of oral gepotidacin (1.5 g or 3.0 g) was effective in 95% of cases in people with urogenital gonorrhoea [152]. This antimicrobial agent is under development in different countries in Europe, North America, and Oceania as a part of a phase III RCT for the evaluation of its efficacy and safety; the control group against a tested group in which patients receive the treatment of choice for this infectious process—a combination of oral doses of azithromycin and intramuscular doses of ceftriaxone [153].

**Solithromycin** is one of “new antibiotics”, member of the ketolide group, such as telithomycin. Both agents interact with the 23S subunit of the 16S subunit of ribosomal RNA, resulting in a blockade of bacterial protein synthesis and thus conferring bacteriostatic activity [154]. Solithromycin, like telithromycin, was designed to be tested in Community-Acquired Bacterial Pneumonia [155]. SOLITAIRE-ORAL [156] and SOLITAIRE IV [157] (Phase III RCTs) demonstrated a good non-inferiority clinical response between solithromycin and moxifloxacin in those with this respiratory infection. Meantime, its potential efficacy for urogenital gonorrhoea began to be tested under SOLITAIRE-U phase III RCT [158]. This last phase III compared the activity of this antimicrobial against the standard combination of IV ceftriaxone + oral azithromycin in two groups of patients. Finally, solithromycin was ousted as a first-line alternative against gonorrhoea because not shown the expected results [158]. However, all this leads to the continued development of new antimicrobials to combat multidrug-resistant strains of NG.

**Corallopyronin A** is an alpha-pyrone with antibacterial activity. This capacity relies on its selective blockade of the RpoB subunit of bacterial RNA polymerase [159,160]. Gram-positive and Gram-negative bacteria such as *Staphylococcus aureus* and *Chlamydia* spp, respectively, are susceptible to its effect, allowing to be tested against NG. The activity on NG has demonstrated to be greater the lower the degree of expression of the Mtr–CDE efflux pumps and vice versa [160].

The **ionophore PBT2** is a chelating agent that, in combination with metal ions such as Zn, has the capacity to cause the reversal of resistance against different multi-resistant pathogens such as methicillin-resistant *S. aureus*, *K. pneumoniae* or *Pseudomonas aeruginosa*, making it a fantastic alternative as a therapeutic agent against multi-resistant strains of NG [161].

In addition, “small” molecules under study are showing significant in vitro antimicrobial activity against NG strains, such as SMT-571 and DIS-73285; however, remaining parameters such as toxicity, safety, and pharmacokinetics/pharmacodynamics need to be evaluated [162,163].

### 6.2. New Perspectives beyond Antimicrobials

There is a pressing need for new treatments that can be used alongside antibiotics or on their own. Cationic antimicrobial peptides (CAMPs) are important defence mechanisms that can fight against a wide range of microorganisms. They are already present in phagocytic granules and can be produced by various cell types, including epithelial cells. However, the histone deacetylases (HDAC) enzymes can reduce the expression of genes that encode CAMPs. This is where HDAC inhibitors (HDACi) come in—they can block the HDACs’ action and increase CAMP expression. The use of natural and synthetic HDACi molecules to boost CAMPs on mucosal surfaces has potential therapeutic applications [164].

*N. gonorrhoeae* and *N. meningitidis* are known to produce glycans, including N-acetylneuraminic acid (Neu5Ac), that mimic host structures. This helps the bacteria to evade host immunity. Neu5Ac inhibits complements by increasing the binding of the complement inhibitor factor H (FH). There are two effective strategies for targeting complement activation in Neisseria. The first involves fusing FH microbial-binding domains to IgG Fc, forming FH18-20/Fc. This fusion protein binds to gonococci and mediates complement-dependent killing in vitro, and it has also shown efficacy in animal models of gonorrhoea bacteraemia. The second strategy involves the incorporation of CMP nonulosonate (CMP–NulO) analogues of sialic acid into LOS, thereby preventing complement inhibition by physiologic CMP-Neu5Ac [165,166].

Temperate phages can help prevent and treat gonorrhoea as they can be used to develop anti-gonococcal vaccines [167] and provide lytic enzymes that target gonococcal bacteria. The NG genome contains nine identified prophages, including the lysogenic filamentous phage Ngoϕ6, which has been shown to be effective against a range of Gram-negative bacteria [135].

Silver nanoparticles are being explored as a potential solution for treating NG [168]. A study shows that they have antimicrobial properties that can reduce the viability of gonococcal bacteria by around 35% when compared to the control group. Moreover, these nanoparticles are non-cytotoxic to human fibroblast epithelial cell lines and have an additional antimicrobial effect when combined with cefmetazole against cefmetazole-resistant strains. Therefore, the use of silver nanoparticles or other nanomaterials as excipients to enhance the effectiveness of current antibiotics against MDR strains requires further investigation [169].

An approach currently under examination is photoinactivation, involving the use of blue light with a 405 nm wavelength, known for its exceptional antimicrobial capabilities. This technique induces photoexcitation of bacterial porphyrins, ultimately generating reactive oxygen species (ROS), which possess potent cytotoxic properties [170].

Several molecules have been explored as potential alternatives to antibiotics. Studies have shown that monocaprin and myritoleic fatty acids can be used as active components to develop new prophylactic products for ophthalmia neonatorum [171]. In addition, certain essential oils have been found to exhibit high bactericidal and anti-biofilm activities against NG [172]. Moreover, the dianion fused to oxazoles, a 3D heterocycle, has demonstrated strong and selective antimicrobial activity against NG [173].

Additionally, monoclonal antibodies could be used to combat complement evasion strategies [165] utilised by NG or be directed against promising vaccine candidates that could be adapted for human use to improve effector function [106].

Finally, anti-virulence therapies (AVTs) inhibit virulence factors to prevent infection. It is hypothesised that targeting non-essential virulence factors reduces the selective pressure for resistance, avoiding antimicrobial resistance (AMR) development. AVT approaches for NG have been recently reviewed by Hill et al. [31].

## 7. Conclusions

Certainly, this review began with a couple of questions, “What are we doing wrong? And more importantly, what might we do?” It is well-established that *N. gonorrhoeae* is an important public health issue due to its virulence and its extraordinary capacity to develop resistance to all the antibiotics used for treatment, as well as the silent transmission or spreading. The asymptomatic cases contribute to the spread by constituting a reservoir, and there is the lack of efficient prevention in third-world countries. We all agree that the optimal management of gonococcal infection involves a synergistic plan of action, from prevention programmes including patient zero, rapid diagnosis to avoid the spreading, detection of AMR to shape the treatment, together with the identification of new therapeutic drugs and vaccine candidates. Regarding this last point, vaccines such as 4CMenB used for invasive meningococcal disease have shown promising results suggesting cross-protection against gonorrhoea and could be a potential support for vaccination strategies for prevention. Finally, machine learning is an innovative tool introduced at the clinical level that allows us to know the spreading and the possible patient zero, as well as to predict resistance and possible new targets and new therapeutics. It is imperative to emphasise the urgency of a global strategy to deal with the explosion of new gonorrhoea cases to avoid a return to the 1800s.

## Figures and Tables

**Figure 1 microorganisms-12-00884-f001:**
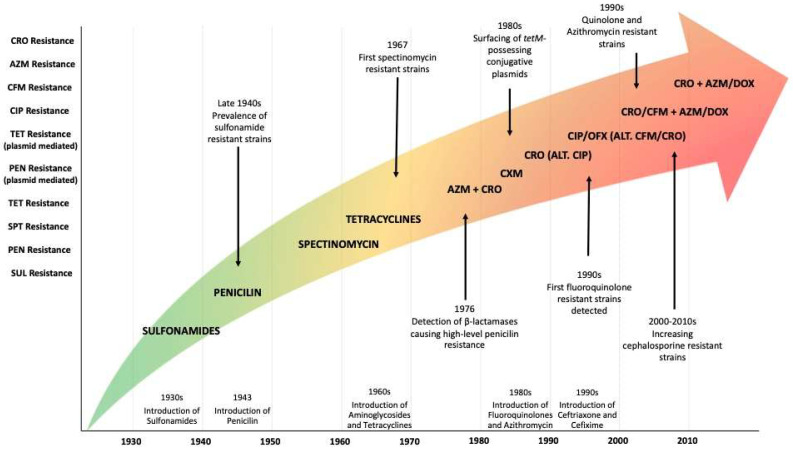
Timeline of developed and recommended antimicrobials paired with the emergence of resistances in *Neisseria gonorrhoeae* during the last century. The centre of the figure shows the recommended treatment regimen at any given moment. The X axis shows the time of introduction of several antimicrobials during the 20th century. The Y axis shows the increasing resistance in NG to said antimicrobials.

**Figure 2 microorganisms-12-00884-f002:**
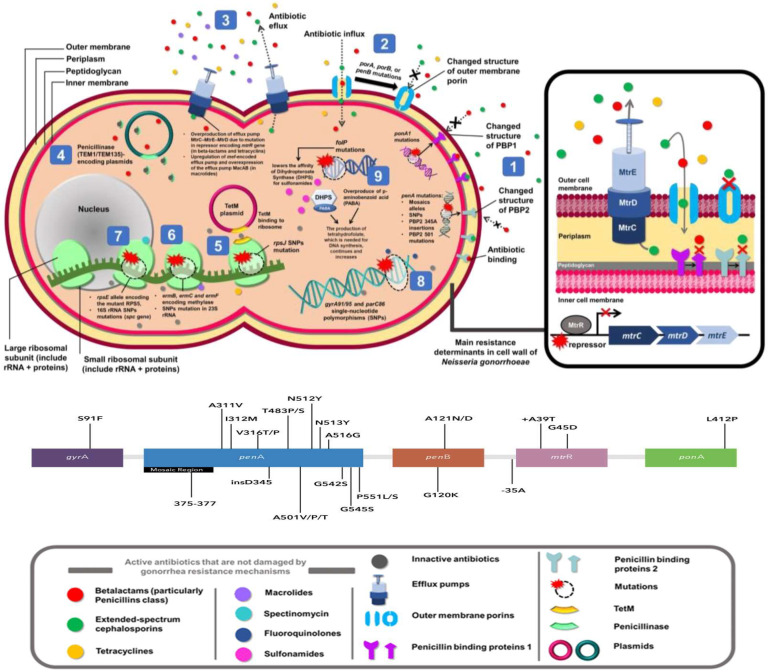
Resistance mechanisms of *N. gonorrhoeae*.

**Table 1 microorganisms-12-00884-t001:** *Neisseria gonorrhoeae* virulence factors, mechanisms of pathogenesis, and role in infection.

Virulence Factor	Pathogenic Mechanisms	Role in Infection
**Lipooligosaccharide** **(LOS)**	Immune evasion	LOS produced by NG can rapidly change its structure due to the production of certain enzymes involved in the biosynthesis of the lacto-N-neotetraose structure [23,24]. Variable oligosaccharide moieties of LOS can mimic host glycosphingolipids [25,26]. LOS can be sialylated, promoting the recruitment of FH to the gonococcal surface and thus rendering the bacteria resistant to serum killing [27,28]. Surface binding of cationic antimicrobial proteins (CAMPs) is reduced by chemical modifications of LOS (PEA-decorated lipid A) [29].
	Adhesion	The LOS of NG can act as a ligand of human receptors, promoting the invasion of host cells [30]
**Type IV pili (Tfp)**	Immune evasion	PilE C-terminal domain undergoes antigenic variation, allowing the bacteria to evade recognition by the human host’s immune cells [31]. *pilC* expression is subject to RecA-independent phase variation (on/off switching) due to frequent frameshift mutations occurring within G tracts located within its signal peptide region [32]. The pilus and porin act in concert to induce calcium fluctuations in the host cell [33]
	Invasion	Altered sequence of the pilE gene in transmigrants implies a variation in the pilin sequence in the transcellular passage of the NG [34].
	Adhesion	Pili attaches to the human mucosal epithelial cells, fallopian tube mucosa, and vaginal epithelial cells, as well as to human polymorphonuclear leukocytes (PMNs; neutrophils). The PilC proteins have been characterised as the major pilus-associated adhesin [35]. NG multiple nonpolar retractile Tfp to elicit adhesive plaque formation in the epithelial cells and requires the protein synthesis and function of the PilT protein [36]. Gonococcal pilus retraction triggers a tight association between gonococcal Opa and host cell receptors [36].
**Opa**	Immune evasion (Phase variation)	A single cell of NG can express none to several Opa proteins, allowing for phase variation that contributes to bacterial resistance to neutrophil clearance [37].
	Adhesion	Opa proteins been shown to interact with CEACAMs on neutrophils and epithelial cells [38,39].
**PorB**	Immune evasion	Suppresses neutrophil oxidative burst and neutrophil apoptosis by binding complement factors C4bp and H [40,41].Delay phagosome maturation and oxidative killing mechanisms [42]. PorB can enter the mitochondria of infected cells via OMVs and form porin channels in the inner membrane. This leads to the release of cytochrome c and other proteins, triggering cell apoptosis [43]. PorB and pili induce calcium transients in host cells, leading to the cleavage of Lamp1 (lysosome-associated membrane protein) by the Neisseria IgA1 protease, and consequently to a reduction in the number of lysosomes in infected cells [33].
**Gonococcal IgA1**	Immune evasion	Reduce mucosal antibody levels by cleaving the hinge region of secretory IgA1. Neisseria IgA1 protease cleaves LAMP1 [44,45].
**Mip**	Immune evasion	Protects bacteria from macrophage killing, probably through mechanisms involving peptidylprolyl cis/trans isomerase (PPIase) activity [46].
**Neisserial Heparin Binding Antigen (NHBA)**	Immune evasion and adhesion	Role in serum resistance, microcolony formation, and adherence to epithelial cells [47].
**OmpA**	Adhesion and invasion	Important for adhesion and invasion in cervical and endometrial cells, as well as entry into macrophages and intracellular survival [48].
**Adhesin Complex Protein (ACP) and SilC**	Immune evasion	Surface-exposed inhibitors of human c-type lysozyme [49,50].
**KatA, cytochrome c, and MsrA and MsrB**	Detoxification and repair of oxidative damage	Catalase KatA and cytochrome c are crucial to Gc defense against ROS. Methionine sulfoxide reductase, MsrA and MsrB, reverses the oxidation of methionine residues in proteins [51].
**RecA and RecN**	Repair oxidative damage	Gc defence against ROS repairing oxidative damage to DNA [51].
**MtrCDE**	Export of antimicrobial components	Efflux pump which enables the bacteria to export various compounds, such as antibiotics, detergents, and antimicrobial peptides, out of the cell [52].
**TbpA, TbpB, LbpA, LbpB, HpuA, FetA, CbpA, ZnuD**	Evasion of Nutritional Immunity	These virulence factors allow bacteria to extract metals like iron and zinc from human innate immune proteins [53].
**MetQ**	Adhesion	Involved in gonococcal adherence to cervical epithelial cells [54].

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
