# Peer review of "Addressing Sexually Transmitted Infections Due to Neisseria gonorrhoeae in the Present and Future"

_microorganisms, 2024, doi:10.3390/microorganisms12050884_

Round 1

Reviewer 1 Report

Comments and Suggestions for Authors

The research primarily addresses the persistent and evolving challenge of Neisseria gonorrhoeae (NG) infections, focusing on resistance mechanisms, prevention programs, treatments, and the potential for vaccine development. It aims to understand why progress in combating this pathogen seems stagnant and explores modern strategies for effective management and control.

This review stands out for its comprehensive examination of the last five years' advances against NG, particularly in new therapeutic alternatives. The paper fills a crucial gap by integrating recent data on resistance patterns, potential vaccines, and improved prevention strategies, offering a fresh perspective on an age-old problem.

Compared to existing literature, this paper adds value by collating and analyzing recent trends in NG resistance and treatment failures, thereby highlighting the urgent need for novel antimicrobial agents, vaccines, and more robust public health strategies. Its focus on up-to-date research and innovative solutions like machine learning for predicting resistance patterns marks a significant contribution to the ongoing battle against sexually transmitted infections (STIs).

While the review methodically compiles existing research, a more detailed analysis of the methodologies used in the primary studies could strengthen its conclusions. Including a meta-analysis of treatments' efficacy and resistance patterns might offer more concrete recommendations. Additionally, further controls like comparing new prevention strategies against established benchmarks could provide clearer insights into their effectiveness.

The conclusions logically follow from the evidence presented, emphasizing the complexity of controlling NG due to its resistance capabilities and asymptomatic transmission. However, while the review thoroughly addresses its main questions through the examination of current knowledge and future directions, it could benefit from more explicit connections between specific experimental results and proposed strategies for NG control.

The references appear appropriate, drawing from a wide range of recent and relevant studies that underline the paper's thoroughness in capturing the current state of NG research. Further scrutiny of the references' quality and their direct relevance to each claim could enhance the paper's credibility.

The inclusion of detailed tables and figures, especially those illustrating resistance mechanisms and the impact of potential vaccines, aids in understanding the complex information presented. Future versions could improve by offering clearer explanations of these visual elements and ensuring all data is up-to-date and accurately referenced.

The methodology section is clear and informative but could be enhanced by explicitly detailing the criteria for selecting studies included in the review. Clarification on the processes for data extraction and analysis would also add rigor.

Comments on the Quality of English Language

The paper's English language quality is generally good, facilitating comprehension. Minor revisions for grammatical precision and clarity could refine the overall presentation.

Author Response

Review 1

The research primarily addresses the persistent and evolving challenge of Neisseria gonorrhoeae (NG) infections, focusing on resistance mechanisms, prevention programs, treatments, and the potential for vaccine development. It aims to understand why progress in combating this pathogen seems stagnant and explores modern strategies for effective management and control.

This review stands out for its comprehensive examination of the last five years' advances against NG, particularly in new therapeutic alternatives. The paper fills a crucial gap by integrating recent data on resistance patterns, potential vaccines, and improved prevention strategies, offering a fresh perspective on an age-old problem.

Compared to existing literature, this paper adds value by collating and analyzing recent trends in NG resistance and treatment failures, thereby highlighting the urgent need for novel antimicrobial agents, vaccines, and more robust public health strategies. Its focus on up-to-date research and innovative solutions like machine learning for predicting resistance patterns marks a significant contribution to the ongoing battle against sexually transmitted infections (STIs).

We are really grateful to reviewer for her/his comments, we really appreciate all her/his suggestions and we hope to address all her/his advises. Thank you

While the review methodically compiles existing research, a more detailed analysis of the methodologies used in the primary studies could strengthen its conclusions. Including a meta-analysis of treatments' efficacy and resistance patterns might offer more concrete recommendations. Additionally, further controls like comparing new prevention strategies against established benchmarks could provide clearer insights into their effectiveness.

Thank for the suggestion. We agree with the reviewer about the meta-analysis of treatment´s efficacy and resistance, so, we have added a new paragraph in 3.4 Resistance-Guided Therapy, line 346-353 together a new references (ref 77-78).

Regarding new prevention strategies, we have added a new paragraph in 390-398. Prevention and Plan of Actions to Overcome Neisseria gonorrhoea, line (ref 89-92).

Although ceftriaxone has proven to be the best performing of the tested injectable drugs in previous meta-analysis, correspondingly, azithromycin was the best oral option77,78. Treatment failures with the standard ceftriaxone dosage are mostly observed in pharyngeal infections, mainly due to pharmacokinetic and pharmacodynamic reasons. In these situations, a higher dose to achieve longer exposure above the MIC may be needed. In recent trials ertapenem has been proposed as an alternative in these situations, but it should not be applied to first-line options to preserve it for other infections caused by MDR Gram-negative bacteria9,14.

The collectives with high-risk sexual behaviors have a higher likelihood of being infected with other STIs. Most of them are patients of PrEP and PEP (Post Exposure Prophylaxis) for the prevention of HIV infection, whom have exponentially increasing over the last decade. Hereby, several studies have been conducted incorporating doxycycline as prophylaxis against other STIs (doxy-PrEP and doxy-PEP)89 because of its high activity against C. trachomatis and T. pallidum and low resistance shown, regardless of well-established resistance patterns for NG and M. genitalium90. Although doxy-PEP can be a excellent STI prevention strategy, more studies are necessary to investigate the long-term impact on resistance profiles91,92.”

The conclusions logically follow from the evidence presented, emphasizing the complexity of controlling NG due to its resistance capabilities and asymptomatic transmission. However, while the review thoroughly addresses its main questions through the examination of current knowledge and future directions, it could benefit from more explicit connections between specific experimental results and proposed strategies for NG control.

Thanks for the recommendation, we appreciate all comments from the reviewer in order to improve the manuscript. The previous paragraph (line 354 and the mentioned line 390-398) are and include more explicit connections experimental vs proposed strategies (ref 79 plus 89-92 ).

Antibiotic stewardship is suggested as a powerful tool to decrease antibiotic pressure when possible, decreasing microbiota exposure and resistance selection79.”

The references appear appropriate, drawing from a wide range of recent and relevant studies that underline the paper's thoroughness in capturing the current state of NG research. Further scrutiny of the references' quality and their direct relevance to each claim could enhance the paper's credibility.

Thanks for the comment, however if the reviewer could give us more information about this input, we will be pleasure to increase the paper´s credibility.

The inclusion of detailed tables and figures, especially those illustrating resistance mechanisms and the impact of potential vaccines, aids in understanding the complex information presented. Future versions could improve by offering clearer explanations of these visual elements and ensuring all data is up-to-date and accurately referenced.

Thanks for the explanation. We have now to expand the legends for table 1 and figure 1, for a better understanding. Also we have accurately reviewed the references. We have added 2 new references in Table1, anyhow the oldest references are usually shown because they are the original and reference manuscript.

The methodology section is clear and informative but could be enhanced by explicitly detailing the criteria for selecting studies included in the review. Clarification on the processes for data extraction and analysis would also add rigor.

Thanks, we have introduced in Introduction more details about the criteria selection for papers. Line 77-81.

“All information has been gathered from relevant articles in PubMed and OVID Medline under the search terms “Neisseria gonorrhoeae” and “antimicrobial resistance” or “epidemiology” or “prevention” or “treatment” or “vaccines” or “virulence factors” or “omics” published from 2016 through 2023. The selection was made based on impact factor of Journal (more than 4) as well as relevance and citations.”

Reviewer 2 Report

Comments and Suggestions for Authors

The manuscript: 

Addressing Sexually Transmitted Infections due to Neisseria gonorrhoeae. Present and Future

Julia Colón Pérez, Rosa Antía Vilarino Fernández, Adrián Domínguez Lago, María Mercedes Treviño Castellano, María Luisa Perez del Molino Bernal, Sandra Sánchez Poza and Eva Torres-Sangiao

Title: corresponds to the content of the article

Introduction: enough; represent the essence of the problem; does not require change

Methodology: meets the requirements of the journal and the branch of knowledge; does not require modification;

Literature: sufficient for the article and does not require additions

Article design: meets the requirements of the journal

Conclusion: the article meets the requirements of the journal and can be published without significant revision

The authors of the article have done a lot of work. The topic is very relevant, especially in the context of growing resistance of Neisseria gonorrhoeae throughout the world. The authors provided a very detailed literature review on the topic under consideration.

This deserves a positive assessment.

There are no comments to this literary review

My only wish is to include a section on the methods for diagnosing gonorrhea that are used today.

Comments on the Quality of English Language

Author Response

 Linked Review 2

The manuscript: 

Addressing Sexually Transmitted Infections due to Neisseria gonorrhoeae. Present and Future

Julia Colón Pérez, Rosa Antía Vilarino Fernández, Adrián Domínguez Lago, María Mercedes Treviño Castellano, María Luisa Perez del Molino Bernal, Sandra Sánchez Poza and Eva Torres-Sangiao

Title: corresponds to the content of the article

Introduction: enough; represent the essence of the problem; does not require change

Methodology: meets the requirements of the journal and the branch of knowledge; does not require modification;

Literature: sufficient for the article and does not require additions

Article design: meets the requirements of the journal

Conclusion: the article meets the requirements of the journal and can be published without significant revision

The authors of the article have done a lot of work. The topic is very relevant, especially in the context of growing resistance of Neisseria gonorrhoeae throughout the world. The authors provided a very detailed literature review on the topic under consideration.

This deserves a positive assessment.

There are no comments to this literary review

My only wish is to include a section on the methods for diagnosing gonorrhea that are used today.

Dear reviewer, thanks for the suggestion. According with that, we have added a new paragraph in section introduction line 42-55. We consider that (to keep the original scheme), a paragraph could be enough to summary the present and future perspectives about the diagnosis. The diagnosis does not change too much in the last decade, and we consider that there are a really good reviews about the topic in case that the reader want to know more about.

“N. gonorrhoeae diagnosis should be motivated by the observation of specific clinical signs or scheduled by regular screenings in high-risk groups. The gonorrhoeae screening is the main key to detect asymptomatic infections as well as co-infections, which are common for C. trachomatis. The current gold-standard for NG diagnosis is nucleic acid amplification tests (NAATs), yet9. NAATs are the recommended diagnostic tools for NG diagnosis in urethral, pharyngeal, and rectal samples, because of their high sensitivity and specificity10. In fact, NG-NAATs are usually incorporated as multiplex panels with syndromic tests, to address the diagnosis of STIs, including other pathogens than NG, such as C. trachomatis, T. pallidum, M. genitalium and T. vaginalis, among others11. In addition, molecular assays are good candidates for point-of-care tests, shortening turnaround time and allowing proper treatment soonest, which has a great impact on transmission and emergence of resistance. However, one limitation of NAATs as daily routine to detect AMR determinants is their high cost. Therefore, conventional culture is still crucial to evaluate NG resistances and must include for diagnostic procedures12.”